# Reflections from a Psychologist Working with Sickle Cell and Thalassaemia Patients during the COVID-19 Pandemic

**DOI:** 10.3390/medicina58091286

**Published:** 2022-09-15

**Authors:** Dede-Kossi Osakonor, Dimitris A. Tsitsikas

**Affiliations:** Haemoglobinopathy Service, Department of Haematology, Homerton University Hospital NHS Foundation Trust, London E9 6SR, UK

**Keywords:** sickle cell disease, thalassaemia, COVID-19, videoconferencing, telehealth, long-term health conditions, psychology

## Abstract

Sickle cell disease and thalassaemia are life-long haematological diseases that can impact the quality of life of patients. This impact on quality of life can require intermittent psychological input throughout the lifespan for management. Managing everyday life during the COVID-19 pandemic could be challenging for people with these health conditions, which could impact their health, their mood and anxiety, their perception of control, and their engagement with their regular healthcare services. This report describes the characteristics of these health conditions and discusses reflections, from a specialist psychology service working with this clinical population, about the impact of COVID-19 on patient engagement with the service. The main aim of this report is to highlight the relevance and usefulness of videoconferencing as a therapy format, suggest implications for further service development and suggest alternate ways of working therapeutically with clients.

## 1. Introduction

The coronavirus disease 2019 (COVID-19) pandemic changed our world; how we interact with others and our environment and how we manage our everyday working and personal lives. Most of the world has experienced some form of quarantine during the pandemic [1]; quarantine being a separation, containment, and restriction of people to prevent the spread of a contagious infection [2]. There have been varied responses in adjusting to an undefined period of living in quarantine and living with the fear of infection, and adjustments have been made to continue to provide healthcare services during this extended period of uncertainty. It is during crises such as the current pandemic that healthcare professionals discover alternate ways of working to continue service provision or develop insights into clinical practice that may continue to be helpful post-pandemic. Telehealth models (telephone and videoconferencing) were found to be helpful pre-COVID-19. In rural Australia circa 2015, oncology services used telehealth to create parity in access to healthcare to mirror the ease of healthcare access and monitoring that was experienced in urban areas in Australia [3]. Benefits of this telehealth service included: cost-effectiveness, conducting patient consultations remotely, remotely supervising and monitoring chemotherapy and other medical treatments, conducting and observing allied health profession sessions and interventions, and conducting psychological therapy [3]. Telehealth and smartphone technology has also been used in creative ways to facilitate adherence to medication and facilitate the adoption of medication regimes by fitting in with everyday life, e.g., a pilot study of electronic-observed-therapy for paediatric sickle cell disease (SCD) patients taking hydroxyurea successfully used smartphones and computers to monitor, remind and record patients taking hydroxyurea with text alerts and video recordings [4]. The use of this interactive and reportedly non-intrusive service led to an increase in hydroxyurea adherence (*p* < 0.01) and 87% of the participants reported that they would continue to use the service [4]; however, no suggestions were provided to explain why 13% of the participants were ambivalent about continuing to use the service.

For the National Health Service (NHS) in the UK, face-to-face consultations were the primary mode of healthcare provision or interaction pre-COVID pandemic. At the start of the COVID-19 pandemic, videoconferencing technology was fast-tracked by the NHS as a tool to facilitate and maintain healthcare provision from a distance during the period of quarantine. Similarly, in the United States, the removal, or an easing of the regulatory barriers in setting up telehealth services was experienced during the start of the COVID-19 pandemic, which led to improved access to healthcare for under-represented populations, e.g., the older adult population [5]. This perspective paper will give a brief overview of both sickle cell and thalassaemia diseases and our specialist psychology service, before discussing some of the reflections made about psychology service provision and patient engagement during the COVID-19 pandemic.

## 2. Sickle Cell Disease and Thalassaemia

Sickle cell disease (SCD) is the most inherited blood disorder in the United Kingdom (UK) [6] and in the United States (US) [7]. It is estimated that 12,500–15,000 people have the disease in the UK [8] and approximately 100,000 people have the disease in the US [7]. This disease is a group of blood disorders that cause the red blood cells to change from a regular round shape to an irregular sickle shape when oxygen concentrations are lowered in the cells [6]. This change in shape reduces the capacity to transport oxygen to other cells in the body and can lead to blockages in blood vessels [7], which can then lead to complications, e.g., multiple organ pain and pain in bones and muscles [6], organ damage, increased vulnerability to infections and an increased likelihood of blood clots and strokes [9]. These complications can lead to multisystemic problems with the hallmark features being acute and persistent pain, frequent hospitalizations, increased risk of infection and reduced life expectancy [6,7]. Treatment can include controlled prescribed analgesics, regular blood transfusions with chelation treatment and a range of prescribed disease-modifying medications, e.g., hydroxyurea/hydroxycarbamide [6].

Similarly, thalassaemia is also a group of inherited blood disorders which is caused by the reduction or absence of haemoglobin in the red blood cells. The hallmark feature of this disease is anaemia which can lead to breathlessness and fatigue, frequent infections, and other complications, i.e., enlarged spleens and stunted growth, which can lead to bone deformation [10]. Treatment can include regular blood transfusions with chelation treatment and controlled prescribed analgesics; persistent pain can also be a feature of this disease due to complications stemming from growth and treatment regimens [10].

## 3. Overview of a Psychology Service for SCD and Thalassaemia

The aim of the service is to help our clients manage life and live well with SCD and thalassaemia. The service caters to young adults and adults from the age of 16, and services a large borough in London, UK. The service has one qualified psychologist and has had additional support from trainee and assistant psychologists in the past. The role involves delivering specialist psychological assessment (including cognitive screening), consultation (including attending hospital ward rounds), formulation and therapeutic interventions (e.g., mindfulness, acceptance and commitment and cognitive-behavioural therapy, systemic approaches, eye-movement desensitization and reprocessing therapy, and brief psychodynamic therapy). We offer advice and consultation on patients’ psychological care to colleagues from other disciplines and work closely with the haemoglobinopathy multidisciplinary team (clinical nurse specialists, junior doctors, haematology consultants, social liaison and welfare benefits officers, and staff nurses) to enable progress in pain management, to improve quality of life, or to reduce unnecessary hospital attendances. Referrals to the psychology service come from external services, e.g., general practitioners (GPs) and the borough’s general psychology service (Improving Access to Psychological Therapies, IAPT), from within the hospital (accident and emergency practitioners, inpatient wards), from the haemoglobinopathy team and by self-referral from the patients themselves.

## 4. Reflection I: Expectation of Increased Demand for the Service Versus Actual Demand

Anecdotal and recorded reports from past health crises or negative global events led us to expect or prepare for increased demand in the need for psychological provision. As discussed earlier, the reason for this expectation was the assumption that the prolonged experience of uncertainty, the fear of infection, the loss of environmental and social freedom and the separation from loved ones could lead to elevated levels of distress and anxiety in the general population. A study from the SARS pandemic in 2002–2003 found that hospital staff who were quarantined were more likely to report anxiety, irritability, detachment from others, difficulties with attention and concentration and experiencing exhaustion [11]. These hospital staff had been working with febrile patients and had been quarantined for nine days, which is not directly comparable to the experience of the general population and to the longevity of the quarantines for the COVID-19 crisis, but the study documented how the experience of quarantine can impact mood, anxiety and behaviour after quarantine, including that the negative effects of quarantine could still be present three years after quarantine ended [12]. Recent reviews and correlational studies report that global levels of depression, anxiety, stress and sleep disturbances have increased since the onset of COVID-19 [13] and that the impact of COVID-19 coincided with elevated levels of stress, anxiety, depressive symptoms and worry and concern for others which impacted sleep, motivation and the experience of physical symptoms in the general population in China [14]. This study also discerned that participants with a lower quality of life experienced a larger impact on their psychological wellbeing. Another study [15] surmised, from their cross-sectional study of the Turkish population, that participants with a chronic health condition were more likely to be psychologically affected by the COVID-19 pandemic.

Consequently, with our psychology service, we expected an increased number of referrals from 1st March 2020 to 30th April 2020 compared to the same period in 2019 (pre-COVID pandemic). This was expected after considering that distress and low mood were more prevalent in the general population and that those with a lower quality of life and chronic health conditions would be more likely to be affected psychologically during the pandemic. However, our service received 6 patient referrals during this two-month period compared to 10 patient referrals during the same period in 2019; the number of referrals was lower than the referrals we received the year before. Whilst this observation of the demands of the service was unexpected, there were a few possible reasons for this observation. Firstly, the SCD and thalassaemia population we manage may have been better able to tolerate the levels of uncertainty that were present at that time. Patients with SCD and thalassaemia may be used to living with or tolerating uncertainty, compared to the general population, because of the unpredictability of their health conditions and because they are used to being hyper-vigilant in monitoring their bodies for symptoms that may require urgent treatment [16]. Secondly, it is possible that the number of referrals was comparably lower because the levels of distress and anxiety in the general population were not as high as expected. It is also possible that fewer referrals were received during this period because there was reduced patient face-to-face contact with health services due to local quarantines and restrictions and there was an increased fear of infection in general. This might have reduced the opportunities to identify patients who might have benefitted from psychological interventions and the opportunities for patients to discuss a potential referral with healthcare professionals. We also compared the number of referrals with those received during the same two-month period in 2021, a year after the initial quarantine period, where we also received six patient referrals. During this period in 2021, it is worth noting that the UK was emerging out of stricter quarantine protocols and there was a slow increase in social mixing allowed outdoors only. Table 1 shows the number of referrals across the same two-month period across three different years.

Comparing these observations with those from other services, there are some similarities and some differences in the trends of referrals. In a paediatric SCD service in Nigeria, there was a significant increase in the uptake of telehealth over a four-month period (April–July 2020): telephone calls increased from 16.5–33.9% and texts to the service increased from 23.3–33.2% [17]. The difference in the context of the contact may explain why this service experienced an increase in telehealth uptake or referrals where our psychology service did not; the need for contact was medical and involved requesting advice about how to manage respiratory distress and other medical presentations, but also involved requesting prescriptions. A multidisciplinary adult haemophilia service in Ireland reported that their clinical contacts (including new patients) were initially lower in April 2020 (*n* = 230) than in April 2019 (*n* = 305) [18]. However, it was also reported that the number of clinical contacts later increased in May 2020 (*n* = 407), which was comparably higher than the number of clinical contacts reported in May 2019 (*n* = 268) and was suggested to be a consequence of the increased integration and optimization of telehealth in their service [18]. When considering our psychology service, we can conclude that the number of referrals to the service remained the same in 2021 because we were still experiencing a quarantine, albeit a less strict one. As mentioned earlier, reduced face-to-face contact may have had an impact on the opportunities to identify patients who might have benefitted from psychological input and may also have reduced the opportunities for patients to self-refer or be aware that they could attend sessions virtually or over the telephone.

During telephone or videocall contacts in 2020 and 2021, we observed that our patients were not more anxious than usual and generally reporting that they were “coping well” or “feeling the same as everyone else”. It could be argued that knowing that most countries across the globe had enforced nationwide quarantines may have been reassuring for this clinical population that experiences continual periods of uncertainty and may isolate at times in order to manage or recover from acute illness. It has been argued that quarantining larger groups of people can help to mitigate some of the stigma and negative consequences of quarantine [2].

## 5. Reflection II: Changes in the Attendance Rate

Our service, like many others in the country and around the world, turned to telehealth to continue service provision during the COVID-19 pandemic. Clinicians [19] have discussed the “Black Swan” for mental health being the turning point for e-health. In the past there has been resistance regarding the use of videoconferencing for continued psychological therapy provision due to challenges with confidentiality, perceived changes in the therapeutic relationship, experiencing a reduced perceptual sense for both the therapist and client, and challenges in candour and managing risk with clients. Evidence supports the effectiveness of internet interventions in managing anxiety and mood disorders [20]. There have been observations around the hastening of the use of telehealth that noticed the use of pragmatic methods to continue service provision during the pandemic [19], where the use of these forums would not otherwise have been considered. In addition, concerns about attendance rates for sessions decreasing because of client unfamiliarity with using videoconferencing or telephone calls may have been unfounded. The benefits of regular telephone contact (the check-in with patients) are known to help reduce anxiety and maintain social contact during pandemics. These telephone sessions do not need to be the same length of time as usual therapy sessions or be as dynamically focussed as face-to-face sessions [21]; brief and solution-focussed telephone sessions can be helpful in their simplicity and directiveness, especially during periods when the general population receive more-than-usual information from social media and media outlets reporting propaganda [2].

A study of 50 outpatients who were offered videoconferencing psychological therapy (rather than waiting for face-to-face sessions) compared to 37 self-referred clients showed that whilst the self-referred clients were more adherent to the components of therapy, there was no difference in the attendance rate for sessions between the two groups [22]. This could imply that we did not need to expect our attendance rates to drop when adapting to videoconferencing and telephone calling, but that we needed to nurture the therapeutic relationship more than usual when using these forums for therapy [22].

Our service documented a rise in the attendance rate from 1 March–30 April 2020. Forty-five patients attended videoconferencing or had telephone contact at scheduled times during this period, compared to thirty-six patients attending face-to-face sessions during the same period in 2019. In 2021, sixty-seven patients attended videoconferencing or had telephone contact during this period. This continued rise in attendance could be explained by the ease with which patients found they could attend their sessions using their smartphones, tablets or computers from the comfort of their homes, compared to travelling to the hospital for their outpatient therapy appointments. These patients can find it challenging to make their outpatient appointments when they have elevated levels of both persistent and acute pain [6], and offering an alternative form of attending appointments may be helpful in maintaining engagement and consistency in attending appointments. In addition, the use of smartphone applications to monitor, be reminded of or receive information about treatment has been found to be acceptable by the younger generation (12–35 years old) of patients with SCD and thalassaemia [23].

A multidisciplinary haemophilia service for adults also reported an improvement in attendance rates by 40% after the introduction of telehealth services [18], for reasons similar to those observed with our clinical population. In addition, frequent telephone contact was found to be an important and flexible way of supporting SCD and thalassaemia patients by a multidisciplinary haematology team in the UK. Telephone support was observed to be helpful by providing accurate health information to patients, by maintaining contact with more vulnerable and at-risk patients, by reminding patients of scheduled treatments (e.g., transfusions and blood tests), and by encouraging patients to maintain social distancing rules [24]. Out of 78 patients, 43 reported that they felt supported by the haematology team during the height of the COVID-19 pandemic (March–May 2020) and 47/78 patients reported that telephone consultations addressed their health concerns [24]. Similar reports of ease of access to services with the integration of telehealth models have been reported in other services [3,4,17,18,25].

Patient satisfaction with the addition of telehealth services is also important in maintaining engagement through clinical contact and in meeting the needs of patients during the pandemic. The use of telehealth services has been shown to significantly reduce hospital admissions and the mean length of hospital stay in SCD and thalassaemia patients compared to a pre-COVID period from 2015–2019 [24]. A survey of 100 haemophilia patients reported that they were less inconvenienced by telehealth consultations (82%) and found this model of their service easy to use (94%) [18]; further, 70% of 1452 rheumatology patients were satisfied with the use of telehealth during the COVID-19 pandemic [26]. The main reasons cited for dissatisfaction with telehealth provision were visual impairment, and the potential risk that comes from fewer clinical examinations and from having limited access to the internet [26]. These are potential issues that need to be considered by all services that use telehealth going forward so that digital poverty and physical use or ability is considered. However, these issues also highlight a need to identify clinical risk more accurately so that the need for clinical examinations is clearer in protocols. This would lead to increased patient and healthcare professional confidence in the integration of telehealth in services.

## 6. Reflection III: Changes in the Focus of Therapy

As discussed earlier, during quarantine periods different stressors can emerge depending on: whether a person quarantines alone or with others, the duration of the quarantine, activity management, fears of infection, being sick, the experience of grief and loss, ease in obtaining supplies, financial constraints and access to information provided about the crisis and how to manage in the crisis [2]. Exploring and addressing these stressors is part of the therapeutic work we do with patients in appointments. We had to “park” past objectives of therapy for most of the clients because the most dominating factor in their lives was how to manage and maintain themselves during the intermittent quarantine periods and manage the uncertainty of future quarantine periods.

The fear of infection is a valid fear for everyone. The population of SCD and thalassaemia have an increased risk of infection due to compromised immune systems, especially after spleen removal, and have an increased risk of complications. The therapy sessions address this by exploring current health fears, health beliefs and behaviours and providing succinct and accurate information about symptomology and what to do for certain physical presentations. The fear of infection also extends to a fear of presenting to hospital for treatment in case COVID-19 is contracted there. It has reportedly been helpful to explore symptomology and discuss the treatment options available with patients while providing reassurances about hospital access, safety and availability, as this was a real concern for many patients. Expectedly, some patients have delayed attending the hospital for acute problems that required hospital intervention because of the fear of contracting an infection in hospital. This can be moderated by checking in, by telephone, with clients about their health status, reassuring those who need to access hospital care but are afraid to and reassuring those who need to know that hospital care is be available to them if they need it. It is also worth noting that the fear of infection could also impact school and work attendance and that options for remote working may improve attendance, confidence and perceptions of safety by mirroring the benefits of telehealth through facilitating work to continue or being able to attend school remotely. Remote working or school attendance could also be helpful in allowing patients the comfort of being at home during increased pain days or days when patients are not feeling as physically able to attend in person.

There has been an increase in the focus on activity management. The change and loss of routine, reduced physical and social activity, loneliness, confinement and being “without purpose” can lead to frustration and boredom during quarantine [2], which can lead to a low mood and the onset of depression [10]. The focus of psychological therapy sessions was to explore patients’ valued activities and reconnect them with these activities, in addition to developing and maintaining a regular routine of waking, being awake, preparing and eating meals, engaging in movement and gentle activity, engaging in social activity and maintaining contact with significant others, completing an orderly task (cleaning or tidying), and sleeping at an appropriate time for them. It has been argued that engaging in these types of activities can combat the feelings of loneliness and loss [27]: loss experienced from not having contact with valued others we do not quarantine with, from changes in regular routine and from focussing inwards unhelpfully. It is argued that loneliness can be transformed to solitude, defined as isolating with peace and tranquillity [27], and that by engaging in these everyday tasks that provide a sense of routine and purpose, people will stay motivated and goal-oriented with a managed mood. In addition, loneliness is known to be a risk factor for impaired health, namely, cardiovascular conditions, fractures and frailty, connective tissue disorders and sensory loss [28], and developing such complications would not be helpful for the management of SCD and thalassaemia. A four-week telehealth cognitive behavioural therapy intervention was found to significantly reduce stress (*p* < 0.01) and depression (*p* < 0.05) in patients with cystic fibrosis and in their caregivers [29]. Levels of anxiety were not impacted by this intervention, and it was suggested that the persistent anxiety was fuelled by the COVID-19 pandemic. An additional explanation could be that the cystic fibrosis patients could possibly naturally report elevated levels of anxiety, compared to the general population, because they, like SCD and thalassaemia patients, may be naturally hypervigilant in order to monitor symptoms and flare ups of symptoms as a disease-management strategy [16].

It is well known that psychological support offered during and after disasters and health crises is known to help mediate the impact of trauma-related stress and is part of the system that maintains and restores emotional resilience and stability. It has been documented that the organizational structure used in China’s experience of the COVID-19 pandemic highlighted the importance of incorporating psychological formulation and assistance throughout the crisis [30,31]. It is important to note that having knowledge of and having access to psychological support can be a protective factor in itself, and it is hoped that our clinical population may benefit from knowing that they can access specialist psychological support as and when they need it.

## 7. Implications and Applications

Some of the changes that have been implemented have been helpful as we continue to navigate the world with outbreaks of COVID-19, but some may also need to be addressed in therapy with our patients, depending on the usefulness of the changes and the impact on everyday life. It may be worth considering that there may be maintained health-behaviour changes after quarantine periods, e.g., hypervigilance and compulsive behaviour (excessive washing of hands, increased checking and vigilance, excessive use of hand sanitiser, etc.), avoidance of crowds and social gatherings, avoidance of busy modes of transport, etc. Time will give us a clearer indication of what the long-term impact of these changes may be and how relevant focussing on or addressing these changes in psychological therapy may be. Checking whether patients understand information that has been provided to them and providing them with additional or more accurate information can be clarifying and reassuring to them. Exploring activity management can also be a helpful strategy for fostering engagement and planning that can help to regulate mood and manage anxiety. It may be important to consider that whilst the demand for our service has increased, the literature does suggest that demand may continue to increase as the pandemic becomes more stable or gets downgraded to an epidemic [12], and as the general population continue to move from survival mode to continued existence.

Videoconferencing and telephone contact or check-ins have been essential and useful tools in enabling us to continue therapeutic engagement with clients and we are continuing to use this medium going forward; not as the sole medium for therapy, but for when clients are less physically able to attend sessions or for routine “check-ins” with clients. Following a “blended” or a hybrid care approach can help to maintain engagement and continuity of care and create flexibility where there was none before [19,24]. These helpful adaptations have also been shown to be helpful for other health professionals who work with this clinical population [24,29] and not only for the psychological provision for SCD and thalassaemia, but also for other health conditions and across health professions [3,5,18]. It may be helpful to note that whilst videoconferencing has been found to be helpful, it can be a tiring and challenging endeavour at times—with internet connection problems, increased vigilance as a psychological practitioner in these sessions, maintaining confidentiality, maintaining professionalism, and staying therapy focussed. We also need to be mindful that telehealth interventions may not be suitable for everyone and that we need to consider how to work with sensory impairment, digital poverty and patients with limited internet access if face-to-face consultations are not favourable or possible. These are uncertain and challenging times that we live in; however, it has been refreshing and humbling to see how fluid the changes that have emerged can be and how versatile and adaptable patients and colleagues have been in working towards maintaining patient engagement and in taking care of each other.

## Figures and Tables

**Table 1 medicina-58-01286-t001:** Number of referrals across different time periods.

Time Period	1 March–30 April 2019	1 March–30 April 2020	1 March–30 April 2021
Type of quarantine	No quarantine	Strict quarantine	Less strict quarantine
Referrals to psychology service	*n* = 10	*n* = 6	*n* = 6

*n* represents the number of SCD and thalassaemia patients referred to the psychology service.

## Data Availability

Not applicable.

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
