# Peer review of "Reflections from a Psychologist Working with Sickle Cell and Thalassaemia Patients during the COVID-19 Pandemic"

_medicina, 2022, doi:10.3390/medicina58091286_

Round 1

Reviewer 1 Report

This is an interesting commentary on the challenges encountered maintaining psychological interventions for adolescents and adults with blood disorders such as SCD or thalassemia in the UK. The author’s experiences are similar to those encountered by my colleagues in the US. I had a number of minor suggestions to improve the manuscript:

1.      Page 3, lines 104-107; perhaps the authors could elaborate on how these referrals are made. For example, if typically coming from healthcare providers, perhaps the reduction reflected fewer visits with healthcare providers and therefore fewer opportunities to make psychology referrals-please clarify?

2.      Page 5, line 235; in discussing videoconferencing for psychology it might be useful to discuss/reference some of the published experience in the US using video conferencing for healthcare visits with SCD patients during the covid pandemic

3.      A brief discussion of the psychological impacts of remote school or remote work (or inability to work in customer facing activities) during the pandemic would be appropriate.

Author Response

Dear reviewer,

Thank you for your feedback. I appreciate the time you took to review the paper and for your useful comments that will help me to improve the quality of the paper so that it can be useful to others. I made amendments that you suggested and hope that they addressed all the points that you made.

1. I have added information about how referrals are made and what the impact of the pandemic would have had on referral routes e.g. lines 98 & 143.

2. I have now referenced other SCD services to show similarities and comparisons when using telehealth e.g. lines  41, 157

3. I briefly mentioned the impact on school and working with this population i.e. line 277

Thank you.

Reviewer 2 Report

Dear authors,

thank you for providing your view on the COVID pandemic and it"s impact on your SCD/thalasemia psychological services. 

It's an interesting topic which certainly needs attention. 

However, I miss a little bit the relation with other, similar services and some concrete results. The presentation is a mixture of narrative overview and a few own impressions. 

Maybe you could add some more substance to the paper? In addition, I miss compassion with similar services and services for non-chronic sick patients. Is there a difference between your cohort and the rest of the population? 

Many thanks

Author Response

Dear reviewer,

Thank you for your feedback. I appreciate the time you took to review the paper and for your useful comments that will help me to improve the quality of the paper so that it can be useful to others. I made amendments that you suggested and hope that they addressed all the points that you made.

- I have added to the introduction to widen the scope and have added more recent and additional relevant references to the paper.

- I have enriched the reflections with reports and observations about telehealth with other health services, e.g. rheumatology, haemophilia, oncology and SCD services in other countries and have critically noted similarities and differences in our observations or reports of use with telehealth e.g. 157, 164, 225, 244, 301.

- I have tried to improve the criticality of the paper and added a table to graphically represent the comparison of the years. Also, I included data from 2021, which I had not used before to show a trend in referrals and in attendances.

Thank you.

Reviewer 3 Report

This article offers some neat observations from a psychologist's perspective working with a vulnerable patient group during the COVID-19 pandemic, namely adolescents and young adults with Sickle Cell Disease and Thalassemia. Overall the insights are noteworthy, however I have a few suggestions below to enhance the overall quality of the manuscript

1) In the first line of the abstract, the authors state 'Sickle Cell Disease and Thalassaemia are life-long haematological diseases that can impact the quality of life of sufferers'. I suggest changing the word 'sufferers' to 'patients' as the word 'sufferers' has negative connotations and, from my own experience, patients dislike this term.

2) In Reflection 1 the authors describe how referrals to the service from March 1st until April 30th actually decreased to 6 referrals compared with 10 in the same period in the preceding year. I feel that this was far too early a time point during the pandemic, especially since the UK government only announced the lockdown on March 23rd. As such it would take a number of weeks for quarantine-related mental health issues to arise and possibly a greater time lag before psychology referrals increased. The authors themselves note in the concluding remarks that demand for psychology services may increase 'when people move from survival mode to continued existence'. Therefore I feel that a later period of comparison e.g. late 2020 or 2021 would be a more suitable comparison than the very early days of the pandemic, when most of the population were in 'survival' mode. In my experience treating SCD and Thalassemia patients, 2022 has been the year I have referred the highest number of patients to Psychology services, suggesting that psychological burdens of the pandemic have increased over time. For all these reasons, I recommend that the authors either change the current timeline for Reflection 1 or add a later time point comparison in addition to the one described here. The latter option may show a different trend over time as the pandemic unfolded.

3) The authors highlight the benefits of using tele-conferencing techniques in their SCD and Thalassemia cohort. To lend strength to their observations, it would be useful to refer to outcomes of tele-medicine in other areas of Haematology which also adopted these techniques during COVID-19 e.g. O'Donovan et al, 'Telehealth for delivery of haemophilia comprehensive care during the COVID-19 pandemic' found e-health was acceptable to patients and healthcare providers across a range of health care services including Psychology, Dentistry and Physiotherapy. Evidence of acceptability across disciplines for other chronic haematological conditions suggests that multi-diciplinary care could also be provided to SCD and thalassemia patients through telemedicine in the future and is not limited to Psyschology alone.

4) The authors make an excellent point in relation to enhanced attendance which 'could be explained by the ease with which patients found they could attend their sessions using their smart phones,  tablets or computers from the comfort of their homes, compared to travelling to the hospital...' especially due to acute/chronic pain experienced by SCD patients. Other factors which help make tele-health more acceptable, particularly in the context of SCD/thalassemia patients is the relatively young age of this cohort (e.g. compared to a geriatric specialty). Moreover, SCD patients have indicated that use of smartphones is very helpful in their care e.g. the use of smartphone applications to assist with adherence and to help with forgetfulness (Fogarty et al, Adherence to hydroxyurea, health related quality of life domains and attitudes towards a smartphone app among Irish adolescents and young adults with sickle cell disease). Adaptability of the traditional healthcare service to include e-health components may improve attendance and overall healthcare engagement of SCD/thalassemia patients. I suggest addition of these points to lend strength to the author's own observations. 

Author Response

Dear reviewer,

Thank you for your feedback. I appreciate the time you took to review the paper and for your useful comments that will help me to improve the quality of the paper so that it can be useful to others. I made amendments that you suggested and hope that they addressed all the points that you made.

- I added additional references to support or critique my observations, including the references that you suggested, Fogarty et al., 2022 and O'Donovan et al., 2020. 

- I included how referrals are made to the psychology service and provided an explanation for how the referral pathway may have been impacted by the pandemic and consequently the referrals made e.g. line 98 & 143.

- You are correct about the outdated term 'sufferers' and I have changed the term to patients to reflect this.

- You are accurate in your observations about the influx of referrals probably increasing as we move through the pandemic. I kept the initial comparison period of March - April 2020, but also included another comparison period of March - April 2021 when the UK was in a more relaxed lock-down to show a trend of what the referral period would be like during the pandemic, rather than as a consequence after the pandemic. I also added this later comparison period to reflection 2 with the attendance rates.

- I have now referenced (to support and critique) other haematology findings with telehealth, but also with oncology, rheumatology and other SCD services e.g. lines 157, 164, 225, 244, 301.

- I have referenced Fogarty et al., 2022, but have also briefly discussed the acceptability of smartphone and tech with younger populations and the advantages of telehealth could also mirror the advantages of remote working and schooling for this population e.g. lines 43, 222, 277.

Thank you.

Round 2

Reviewer 2 Report

Dear authors,

thank you for re-submitting your manuscript.

I have the feeling most of my comments were not addressed.

Kindly do so and/or reply.

Best regards 

Author Response

Dear reviewer,

Thank you for your comments. I will discuss with the team.

Best wishes.